

# Evolution of developmental sequences in lepidosaurs

Tomasz Skawiński and Bartosz Borczyk

Department of Evolutionary Biology and Conservation of Vertebrates, University of Wroclaw, Wrocław, Poland

## ABSTRACT

**Background**. Lepidosaurs, a group including rhynchocephalians and squamates, are one of the major clades of extant vertebrates. Although there has been extensive phylogenetic work on this clade, its interrelationships are a matter of debate. Morphological and molecular data suggest very different relationships within squamates. Despite this, relatively few studies have assessed the utility of other types of data for inferring squamate phylogeny.

**Methods**. We used developmental sequences of 20 events in 29 species of lepidosaurs. These sequences were analysed using event-pairing and continuous analysis. They were transformed into cladistic characters and analysed in TNT. Ancestral state reconstructions were performed on two main phylogenetic hypotheses of squamates (morphological and molecular).

**Results**. Cladistic analyses conducted using characters generated by these methods do not resemble any previously published phylogeny. Ancestral state reconstructions are equally consistent with both morphological and molecular hypotheses of squamate phylogeny. Only several inferred heterochronic events are common to all methods and phylogenies.

**Discussion**. Results of the cladistic analyses, and the fact that reconstructions of heterochronic events show more similarities between certain methods rather than phylogenetic hypotheses, suggest that phylogenetic signal is at best weak in the studied developmental events. Possibly the developmental sequences analysed here evolve too quickly to recover deep divergences within Squamata.

Corresponding author
Tomasz Skawiński,
tomasz.skawinski@uwr.edu.pl,
tomasz.skawinski@o2.pl

## INTRODUCTION

With over 10,000 species, Squamata (lizards, snakes and amphisbaenians) are one of the most species-rich extant tetrapod lineages (*Uetz, Freed & Hošek, 2016*). However, our understanding of their evolutionary history is confounded by the conflict between phylogenetic hypotheses based on morphology and molecular data (e.g., *Losos, Hillis & Greene, 2012*). Morphological analyses suggest that the first divergence within Squamata was between Iguania (iguanas, agamas, chameleons and kin) and Scleroglossa (all other lizards and snakes) (e.g., *Estes, de Queiroz & Gauthier, 1988*; *Conrad, 2008*; *Gauthier et al., 2012*), while molecular studies indicate that iguanians are highly derived lizards, closely related to anguimorphs (e.g., monitor lizards) and snakes, and that limbless dibamids or

gekkotans (geckos and kin, sometimes also including dibamids) are the first-diverging branch of squamates (e.g., *Townsend et al., 2004*; *Vidal & Hedges, 2005*; *Wiens et al., 2010*; *Wiens et al., 2012*; *Pyron, Burbrink & Wiens, 2013*). Increasing the number of taxa and characters in these analyses has not led to an improvement of our understanding of squamate phylogeny, but rather has only increased the discordance between the hypotheses based on those two lines of evidence. Combined morphological and molecular analyses (e.g., *Wiens et al., 2010*; *Reeder et al., 2015*) generally favour the molecular topology (but see *Lee, 2005*). However, some authors argue that molecular data may not be ideal for resolving the higher-level phylogeny of squamates because of the large genetic distance between squamates and their closest living relative—the tuatara (*Sphenodon punctatus*), the only extant rhynchocephalian—and thus the only reasonable proximal outgroup to Squamata in phylogenetic analyses (*McMahan et al., 2015*). Despite numerous publications on this subject (*Gauthier et al., 2012*; *Losos, Hillis & Greene, 2012*; *Reeder et al., 2015*), the debate continues and still new approaches to the problem are being taken (*McMahan et al., 2015*; *Harrington, Leavitt & Reeder, 2016*; *Pyron, 2017*).

Developmental data may be useful for phylogenetic inference (e.g., *Laurin & Germain, 2011*) but they rarely have been used in squamate phylogenetics. Notable exceptions are the studies of *Maisano (2002)* and *Werneburg & Sánchez-Villagra (2015)*, using ossification sequences. The former found that these sequences are useful for determininig relatively shallow divergences but failed to recover deeper nodes, possibly because of their high rate of evolution (*Maisano, 2002*). *Werneburg & Sánchez-Villagra (2015)* found that developmental data were most congruent with the close relationship between snakes and varanids, as postulated by some morphological studies (e.g., *Lee, 1997*) but also some combined morphological and molecular analyses (e.g., *Lee, 2005*). Sequences of other developmental traits were studied by *Andrews, Brandley & Greene (2013)* but the authors regarded relationships of squamates as ''well defined'' and reconstructed the ancestral states only on the molecular topologies. Moreover, their study did not consider the tuatara, a taxon critical in studying lepidosaur evolution. We attempt to supplement their data with the developmental sequence of the tuatara and reconstruct ancestral states using both molecular and morphological topologies. We also assess phylogenetic utility of timing of organogenesis using several different methods.

## MATERIALS & METHODS

### Character construction and cladistic analyses

Developmental sequences of 20 characters in 21 species representing most major squamate lineages (Tables 1 and 2) were obtained from *Andrews, Brandley & Greene (2013)*. Developmental sequences of seven other squamate species were taken from the literature (Table 1). The developmental sequence of the tuatara was compiled from *Dendy (1899)* and *Sanger, Gredler & Cohn (2015)* (see also *Moffat, 1985*). These sequences were transformed into continuous characters, where the first event has a value of 0, and the last one—a value of 1 (*Germain & Laurin, 2009*; *Laurin & Germain, 2011*). These values constituted the basis for cladistic characters, which were created following *Werneburg & Sánchez-Villagra*

**Table 1** Species included in this study, their taxonomic position and sources of information on their development.

| Species | Higher taxon | Source |
|---|---|---|
| *Sphenodon punctatus* (Gray, 1842) | Rhynchocephalia: Sphenodontidae | *Dendy (1899)*, *Moffat (1985)* and *Sanger, Gredler & Cohn (2015)* |
| *Amalosia lesueurii* (Duméril & Bibron, 1836) | Gekkota: Diplodactylidae | *Andrews, Brandley & Greene (2013)* |
| *Strophurus williamsi* (Kluge, 1963) | Gekkota: Diplodactylidae | *Andrews, Brandley & Greene (2013)* |
| *Eublepharis macularius* (Blyth, 1854) | Gekkota: Eublepharidae | *Andrews, Brandley & Greene (2013)*[a] |
| *Tarentola annularis* (Geoffroy Saint-Hilaire, 1827) | Gekkota: Phyllodactylidae | *Khannoon (2015)* |
| *Chondrodactylus turneri* (Gray, 1864) | Gekkota: Gekkonidae | *Andrews, Brandley & Greene (2013)* |
| *Gehyra variegata* (Duméril & Bibron, 1836) | Gekkota: Gekkonidae | *Andrews, Brandley & Greene (2013)* |
| *Mabuya* sp. | Scincoidea: Scincidae | *Andrews, Brandley & Greene (2013)* |
| *Calyptommatus sinebrachiatus* Rodrigues, 1991 | Lacertiformes: Gymnophthalmidae | *Andrews, Brandley & Greene (2013)* |
| *Nothobachia ablephara* Rodrigues, 1984 | Lacertiformes: Gymnophthalmidae | *Andrews, Brandley & Greene (2013)* |
| *Zootoca vivipara* (Lichtenstein, 1823) | Lacertiformes: Lacertidae | *Andrews, Brandley & Greene (2013)* |
| *Python sebae* (Gmelin, 1789) | Serpentes: Pythonidae | *Boughner et al. (2007)* |
| *Thamnophis sirtalis* (Linnaeus, 1758) | Serpentes: Colubridae | *Andrews, Brandley & Greene (2013)* |
| *Boaedon fuliginosus* (Boie, 1827) | Serpentes: Lamprophiidae | *Boback, Dichter & Mistry (2012)* |
| *Vipera aspis* (Linnaeus, 1758) | Serpentes: Viperidae | *Andrews, Brandley & Greene (2013)* |
| *Varanus rosenbergi* Mertens, 1957 | Anguimorpha: Varanidae | *Andrews, Brandley & Greene (2013)* |
| *Varanus indicus* (Daudin, 1802) | Anguimorpha: Varanidae | *Gregorovicova et al. (2012)* |
| *Varanus panoptes* Storr, 1980 | Anguimorpha: Varanidae | *Werneburg, Polachowski & Hutchinson (2015)* |
| *Iguana iguana* (Linnaeus, 1758) | Iguania: Pleurodonta: Iguanidae | *Lima (2015)* |
| *Uta stansburiana* Baird & Girard, 1852 | Iguania: Pleurodonta: Phrynosomatidae | *Andrews, Brandley & Greene (2013)* |
| *Anolis sagrei* Duméril & Bibron, 1837 | Iguania: Pleurodonta: Dactyloidae | *Andrews, Brandley & Greene (2013)* |
| *Liolaemus gravenhorsti* (Gray, 1845) | Iguania: Pleurodonta: Liolaemidae | *Andrews, Brandley & Greene (2013)* |
| *Liolaemus tenuis* (Duméril & Bibron, 1837) | Iguania: Pleurodonta: Liolaemidae | *Andrews, Brandley & Greene (2013)* |
| *Tropidurus torquatus* (Wied-Neuwied, 1820) | Iguania: Pleurodonta: Tropiduridae | *Py-Daniel et al. (2017)* |
| *Chamaeleo calyptratus* Duméril & Duméril, 1851 | Iguania: Acrodonta: Chamaeleonidae | *Andrews, Brandley & Greene (2013)* |
| *Furcifer lateralis* (Gray, 1831) | Iguania: Acrodonta: Chamaeleonidae | *Andrews, Brandley & Greene (2013)* |
| *Pogona vitticeps* (Ahl, 1926) | Iguania: Acrodonta: Agamidae | *Andrews, Brandley & Greene (2013)* |
| *Calotes versicolor* (Daudin, 1802) | Iguania: Acrodonta: Agamidae | *Andrews, Brandley & Greene (2013)* |
| *Agama impalearis* Boettger, 1874 | Iguania: Acrodonta: Agamidae | *Andrews, Brandley & Greene (2013)* |

Notes.

[a] *Wise, Vickaryous & Russell (2009)* presented slightly different developmental table for *Eublepharis macularius* but we used data from *Andrews, Brandley & Greene (2013)*, as they span the whole development.

*(2015)*—values between 0 and 0.09 were coded as 0, between 0.1 and 0.19 were coded as 1, and so on. The missing data were coded as unknown (?), while limb characters in snakes were coded as inapplicable (-). Cladistic analyses employing these characters were conducted in TNT v. 1.1 (*Goloboff, Farris & Nixon, 2003*; *Goloboff, Farris & Nixon, 2008*), using the traditional search option, with 10 replications of Wagner trees. These trees were held in RAM and subjected to tree bisection reconnection, holding 10 trees per replicate. In the first analysis, all characters were unordered (non-additive), and in the second one, all were ordered (additive) (see *Werneburg & Sánchez-Villagra, 2009*; *Laurin & Germain, 2011*). In both analyses, *Sphenodon* was used as the outgroup.

**Table 2  Developmental events used in this study.** From *Andrews, Brandley & Greene (2013)*.

| Number | Event |
|--------|-------|
| 1 | Primary optic vesicle |
| 2 | Otic placode |
| 3 | Allantois bud (small thick-walled out-pouching) |
| 4 | Torsion complete |
| 5 | Secondary optic vesicle |
| 6 | Hyomandibular slit |
| 7 | Allantois vesicle (thin-walled bag) |
| 8 | Choroid fissure open (horseshoe-shaped) |
| 9 | Limb ridge |
| 10 | Allantois contacts chorion (allantois flattened above embryo like umbrella) |
| 11 | Maximum pharyngeal slits |
| 12 | Limb Apical Ectodermal Ridge (AER) |
| 13 | Hemipenal buds form on cloacal lip |
| 14 | Three-segmented limb (stylo-, zeugo-, autopodium) |
| 15 | Jaw initiated |
| 16 | Eyelid forms as a thin ribbon-like sheet of tissue overlapping the eyeball |
| 17 | Pharyngeal slits closed |
| 18 | Digits differentiated in limb paddle |
| 19 | Jaw complete; mandible meets tip of maxilla |
| 20 | Scale anlagen visible |

Another set of cladistic characters was created using the event-pairing method (*Smith, 1997*; *Velhagen Jr, 1997*; *Jeffery et al., 2002a*; *Jeffery et al., 2005*). Comparing 20 developmental events in 29 species resulted in 190 event pairs. These characters were analysed in the same way as continuous characters.

With these cladistic characters and files with both molecular and morphological topology in memory, Templeton test (*Templeton, 1983*) was performed in TNT (using a script written by Alexander Schmidt-Lebuhn: https://www.anbg.gov.au/cpbr/tools/templetontest.tnt). Four replications were conducted: using either ordered or unordered characters; and employing continuous or event-paired characters.

## Ancestral state reconstruction and heterochronic events

Reconstruction of ancestral states was performed in Mesquite v. 3.2 (*Maddison & Maddison, 2017*). Developmental sequences were mapped on two competing phylogenetic hypotheses of lepidosaurs—first one, from *Pyron, Burbrink & Wiens (2013)*, using seven nuclear and five mitochondrial genes, and the second one, from *Gauthier et al. (2012)*, the largest morphological analysis to date. Ancestral states were reconstructed using both maximum parsimony and maximum likelihood for event-paired data and square-changed parsimony for continuous data. The branch length may have a significant effect on reconstruction of ancestral states (e.g., *Andrews, Brandley & Greene, 2013*; *Boyd, 2015*), so analyses using maximum likelihood and square-changed parsimony were performed on both molecular and morphological topologies. In the first analysis, all branches were given an equal

**Table 3  Calibration points for the fossil time-calibrated analyses.** See the 'Materials & Methods' section for details.

| Taxon | Age | References | Notes |
|---|---|---|---|
| Sauria | 256 Ma | *Ezcurra, Scheyer & Butler (2014)* and *Ezcurra (2016)* | |
| Rhynchocephalia | 238 Ma | *Jones et al. (2013)* | |
| Iguania | 105 Ma (99 + 3 + 3) | *Daza et al. (2016)* | Much older, Jurassic, fossils may represent iguanians (e.g., *Evans, Prasad & Manhas, 2002*) but their systematic position is ambiguous (e.g., *Jones et al., 2013*). |
| Acrodonta | 102 Ma (99 + 3) | *Daza et al. (2016)* | |
| Chamaeleonidae | 99 Ma | *Daza et al. (2016)* | |
| Agamidae | 99 Ma | *Daza et al. (2016)* | |
| *Chamaeleo* | 13 Ma | *Bolet & Evans (2014)* | |
| Tropiduridae | ca. 15 Ma | *Conrad, Rieppel & Grande (2007)* | |
| Iguanidae | 56 Ma | *Nydam (2013)* | |
| *Anolis* | 20 Ma | *Sherratt et al. (2015)* | |
| Gekkota | 150 Ma | *Gauthier et al. (2012)* and *Caldwell et al. (2015)* | See also *Daza, Bauer & Snively (2014)* |
| Gekkonidae | 15 Ma | *Daza, Bauer & Snively (2014)* | |
| Diplodactylidae | 20 Ma | *Daza, Bauer & Snively (2014)* | |
| Serpentes | 167 Ma | *Caldwell et al. (2015)* | |
| Pythonidae | 35 Ma | *Head (2015)* | |
| Colubridae | 31 Ma | *Head, Mahlow & Müller (2016)* | |
| Lamprophiidae | 17 Ma | *Head, Mahlow & Müller (2016)* | Based on the elapid *Naja romani* (*Head, Mahlow & Müller 2016*). |
| Viperidae | 20 Ma | *Head, Mahlow & Müller (2016)* | |
| Anguimorpha | 145 Ma | *Head (2015)* and *Caldwell et al. (2015)* | |
| Lacertiformes | 99 Ma | *Daza et al. (2016)* | |
| Gymnophthalmidae | 66 Ma | *Venczel & Codrea (2016)* | Gymnophthalmid fossils are currently unknown (*Nydam & Caldwell, 2015*) but teiids are universally accepted as gymnophthalmid sister group, so the oldest known teiid is used to provide a calibration point for gymnophthalmids in the analyses. |
| Scincoidea | 150 Ma | *Evans & Chure (1998)* and *Gauthier et al. (2012)* | See also *Conrad (2008)* and *Tałanda (2016)*–regardless of that, the oldest known scincoids seem to be Late Jurassic in age. |

length (=1), while in the second, the branch lengths were calibrated to reflect the fossil record of a given group. The oldest-known fossil of a total group was used to calibrate the tree rather than that of a crown group (Table 3). Only fossils unquestionably placed within a given group were included. When the fossil record of a group was unknown (mostly in relatively recently diverged species), the branch length was set, arbitrarily, as 3. Square-changed parsimony reconstruction using continuous data was performed using

root node reconstruction in PDAP:PDTREE module of Mesquite (*Midford, Garland Jr & Maddison, 2011*). This module calculates 95% confidence intervals (*Garland Jr & Ives, 2000*) for each character of a hypothetical ancestor of all taxa included in a tree (in this case, ancestral lepidosaur). A statistically significant heterochronic event occurs when a value of character state of a given taxon is beyond the confidence interval. In the second analysis, *Sphenodon* was pruned from the tree, and reconstruction was made for the ancestral squamate and compared to the values of terminal taxa.

Event-pair synapomorphies were mapped on both topologies using synapomorphy mapping in TNT. These synapomorphies were subjected to event-pair cracking, following the procedure described in detail by *Jeffery et al. (2002a)*. Only deviations from their methods are described below. Clades supported by only one event-pair synapomorphy, two synapomorphies involving four different events and so on were excluded because the number of developmental changes was insufficient for determining the background pattern and heterochronies. In the ordered dataset, when degree of change was ambiguous (e.g., from 0 to 1 or 2), a mean was taken (in this example, 1.5). Characters in which the direction of change could not be unambiguously reconstructed (i.e., from 1 to 0 or 2) were excluded from further analysis. This should not have significant effect on the analysis, as there was only a few such characters (Tables S1–S8). Only events with total relative change (TRC) beyond the 95% confidence interval calculated for the mean TRCs at a given node were regarded as heterochronic. This is more a conservative approach than the one taken by *Jeffery et al. (2002a)* but will make the analysis more comparable to the continuous analysis described above.

## RESULTS

### Cladistic analyses

Cladistic analyses conducted using the transformed continuous data generated trees that are not similar to trees obtained in either morphological or molecular analyses. Analysis using unordered characters yielded 214 most parsimonious trees (MPT; tree length = 109, consistency index = 0.560, retention index = 0.628), the strict consensus tree of which is almost completely unresolved. This analysis failed to recover clades of very closely related species such as *Liolaemus* (Fig. 1A). When all characters were ordered, it resulted in 174 most parsimonious trees (TL = 133, CI = 0.459, RI = 0.625). The strict consensus tree is mostly unresolved—the only groups that were monophyletic in all MPTs are *Liolaemus*, *Tropidurus* + *Strophurus*, *Calyptommatus* + *Anolis* and a clade including *Uta*, *Agama*, *Furcifer*, *Mabuya*, *Gehyra*, *Chamaeleo* and *Zootoca*. A 50% majority rule tree does not resemble published morphological or molecular phylogenies (Fig. 1B).

Similar to the continuous dataset, the event-paired data did not result in a topology matching any previously published phylogeny. Analysis of unordered characters generated 10 MPTs (TL = 185, CI = 0.530, RI = 0.552). In the strict consensus tree *Furcifer* and *Varanus indicus* are in trichotomy with the clade including all other squamates. This clade is divided into a group containing seven species of iguanians, gekkotan *Strophurus*, snake *Thamnophis*, scincoid *Mabuya* and lacertiform *Zootoca*, and the second group to which all

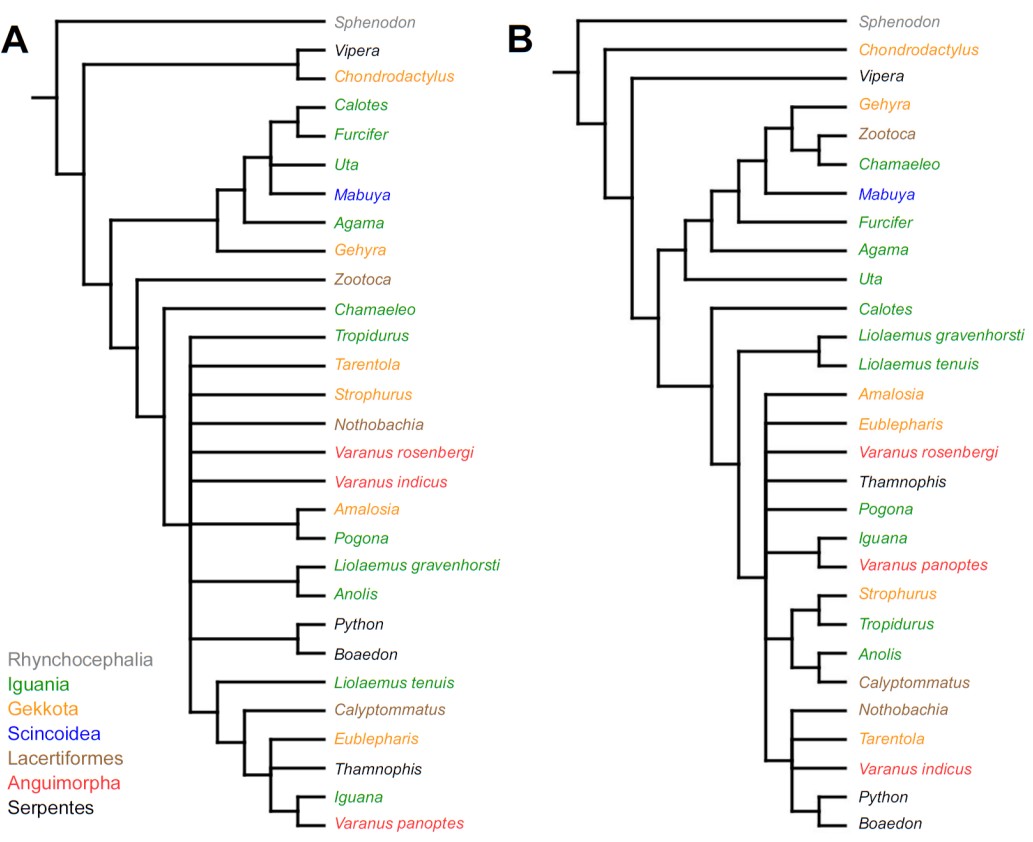

**Figure 1** **Results of the cladistic analysis using characters from the continuous analysis.** Strict consensus tree. (A) Unordered characters; TL = 109, CI = 0.560, RI = 0.628. (B) Ordered characters; TL = 133, CI = 0.459, RI = 0.625. Colour represents clade to which given species belongs.

other squamates belong (Fig. 2A). Analysis using ordered characters yielded 16 MPTs (TL = 220, CI = 0.464, RI = 0.599). The strict consensus tree is poorly resolved but excluding *Varanus indicus* from it significantly improves resolution. After this, squamates are divided into two clades—the first one includes eight species of iguanians, *Thamnophis* and *Mabuya*, while the second group includes all other squamates (Fig. 2B).

Mapping of continuous characters indicates slight differences in tree length between morphological and molecular topologies. With all branches being assigned equal length (=1), the former is 1.49630768 steps long and the latter—1.51610078. With the fossil-calibrated tree, the morphological topology is 0.19257679 steps long and molecular—0.17638729. Mapping of unordered event-paired characters gives the molecular topology a length of 250 steps and the morphological—252 steps. With ordered characters, the molecular topology is 322 steps long, while the morphological is 327 steps long.

Neither replication of the Templeton test detected any statistically significant differences between morphological and molecular phylogenies under both present continuous and event-paired character datasets ($p > 0.05$ in all cases).

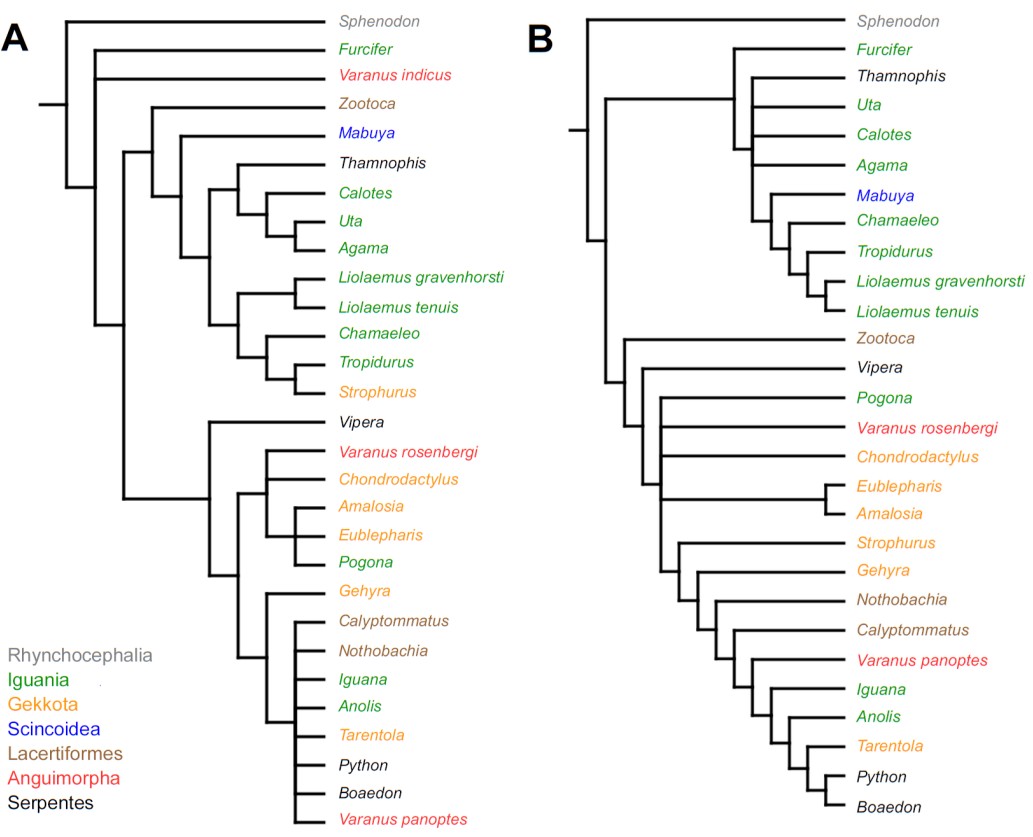

**Figure 2** **Results of the cladistic analysis using characters from event-pairing.** Strict consensus tree (in B after excluding *Varanus indicus*). (A) Unordered characters; TL = 185, CI = 0.530, RI = 0.552. (B) Ordered characters; TL = 220, CI = 0.464, RI = 0.559. Colour represents clade to which given species belongs.

## Developmental diagnoses

There are several event-pair synapomorphies diagnosing some higher-level taxa (i.e., family-level clades or higher). However, at least some of these groups are represented by only a few members (e.g., Anguimorpha, Scincoidea), so these apomorphies may in fact diagnose less inclusive clades (Table 4).

## Heterochronic events

Inferred heterochronic events show more consistency between given methods than between phylogenies (e.g., event-paired data for morphological phylogeny are more similar to event-paired data for molecular topology than to continuous data for morphological tree). Only a few of these events are common to all methods and phylogenies (Figs. 3–14).

## DISCUSSION

Developmental cladistic characters failed to recover topology similar to those based on other data (i.e., molecular or morphological). This was also found in similar studies (*Maisano, 2002*; *Werneburg & Sánchez-Villagra, 2009*; *Werneburg & Sánchez-Villagra, 2015*). This may be a consequence of uneven sampling of different squamate clades in the present

**Table 4 Event-paired developmental synapomorphies of higher-level squamate clades.** Asterisk denotes synapomorphies present only in analysis using ordered characters, while plus denotes synapomorphies present only in analysis employing unordered characters.

| Clade | Synapomorphies |
|---|---|
| **(a) Molecular phylogeny** | |
| Gekkota except Diplodactylidae | (1) pharyngeal slits closed simultaneous with three-segmented limb* |
| Unidentata | (1) secondary optic vesicle simultaneous with allantois bud, (2) hyomandibular slit not earlier than allantois bud |
| Scincoidea (*Mabuya*) | (1) hyomandibular slit later than secondary optic vesicle, (2) allantois vesicle earlier than torsion completion, (3) allantois contacts chorion simultaneous with torsion completion, (4) allantois contacts chorion simultaneous with hyomandibular slit, (5) allantois contacts chorion earlier than choroid fissure open, (6) allantois contacts chorion earlier than limb ridge*, (7) pharyngeal slits closed later than eyelid forms as a thin ribbon-like sheet of tissue overlapping the eyeball* |
| Gymnophthalmidae | (1) jaw initiated simultaneous with maximum pharyngeal slits, (2) jaw initiated earlier than hemipenal buds form on cloacal lips, (3) pharyngeal slits closed simultaneous with hemipenal buds form on cloacal lips*, (4) pharyngeal slits closed simultaneous with three-segmented limb*, (5) jaw completion simultaneous with digits differentiated in the limb paddle |
| Toxicofera | (1) secondary optic vesicle later than allantois bud, (2) allantois vesicle simultaneous with secondary optic vesicle* |
| Serpentes | (1) pharyngeal slits closed no later than hemipenal buds form on cloacal lips*, (2) pharyngeal slits closed earlier than eyelid form as thin ribbon-like sheet of tissue* |
| *Thamnophis* + *Vipera* | (1) jaw initiated later than hemipenal buds form on cloacal lips*, (2) eyelid form as thin ribbon-like sheet of tissue simultaneous with jaw initiated |
| *Varanus rosenbergi* + *V. panoptes* | (1) pharyngeal slits closed simultaneous with three-segmented limb |
| Iguania | (1) limb ridge later than choroid fissure open* |
| Acrodonta | (1) allantois vesicle simultaneous with torsion completion + |
| Chamaeleonidae | (1) allantois contacts chorion later than limb ridge* |
| *Agama* + *Calotes* | (1) jaw initiated later than hemipenal buds form on cloacal lips*, (2) jaw initiated later than three-segmented limb*, (3) pharyngeal slits closed simultaneous with jaw initiated* |
| Pleurodonta excluding *Tropidurus* | (1) pharyngeal slits closed simultaneous with three-segmented limb* |
| *Liolaemus* | (1) jaw initiated simultaneous with three-segmented limb+, (2) pharyngeal slits closed earlier than jaw initiated |
| **(b) Morphological phylogeny** | |
| Iguania | (1) hyomandibular slit later than allantois bud* |
| Pleurodonta | (1) pharyngeal slits closed simultaneous with three-segmented limb*, (2) pharyngeal slits closed earlier than eyelid forms as thin ribbon-like sheet of tissue* |
| *Liolaemus* | (1) allantois bud earlier than otic placode, (2) secondary optic vesicle earlier than otic placode, (3) secondary optic vesicle simultaneous with allantois bud+, (4) hyomandibular slit simultaneous with torsion completion*, (5) hyomandibular slit later than secondary optic vesicle*, (6) choroid fissure open simultaneous with otic placode, (7) choroid fissure open earlier than allantois vesicle, (8) jaw initiated simultaneous with three-segmented limb+, (9) pharyngeal slits closed earlier than three-segmented limb*, (10) pharyngeal slits closed earlier than jaw initiated |
| Acrodonta | (1) allantois vesicle simultaneous with torsion completion+ |
| *Agama* + *Pogona* | (1) eyelid forms as thin ribbon-like sheet of tissue simultaneous with jaw initiated* |
| Chamaeleonidae | (1) allantois contacts chorion later than limb ridge* |

*(continued on next page)*

**Table 4** (*continued*)

| Clade | Synapomorphies |
|---|---|
| Scleroglossa | (1) torsion completion simultaneous with allantois bud*, (2) hyomandibular slit simultaneous with torsion completion |
| Gekkota except Diplodactylidae | (1) pharyngeal slits closed simultaneous with three-segmented limb* |
| *Varanus rosenbergi + V. panoptes* | (1) pharyngeal slits closed simultaneous with three-segmented limb |
| Serpentes | (1) pharyngeal slits closed earlier than eyelid forms as a thin ribbon-like sheet of tissue* |
| Scincomorpha | (1) allantois bud simultaneous with otic placode+, (2) secondary optic vesicle simultaneous with otic placode |
| Scincoidea (*Mabuya*) | (1) torsion completion later than allantois bud*, (2) secondary optic vesicle earlier than torsion completion*, (3) hyomandibular slit later than allantois bud*, (4) hyomandibular slit later than secondary optic vesicle*, (5) allantois vesicle earlier than torsion completion, (6) allantois vesicle earlier than hyomandibular slit, (7) allantois contacts chorion simultaneous with torsion completion, (8) allantois contacts chorion simultaneous with hyomandibular slit, (9) allantois contacts chorion earlier than choroid fissure open, (10) allantois contacts chorion earlier than limb ridge, (11) jaw initiated simultaneous with three-segmented limb+ |
| Gymnophthalmidae | (1) jaw initiated simultaneous with maximum pharyngeal slits, (2) jaw initiated earlier than hemipenal buds form on cloacal lips, (3) jaw initiated earlier than three-segmented limb*, (4) pharyngeal slits closed simultaneous with hemipenal buds form on cloacal lips*, (5) pharyngeal slits closed simultaneous with three-segmented limb*, (6) jaw completion simultaneous with digits differentiated in limb paddle |

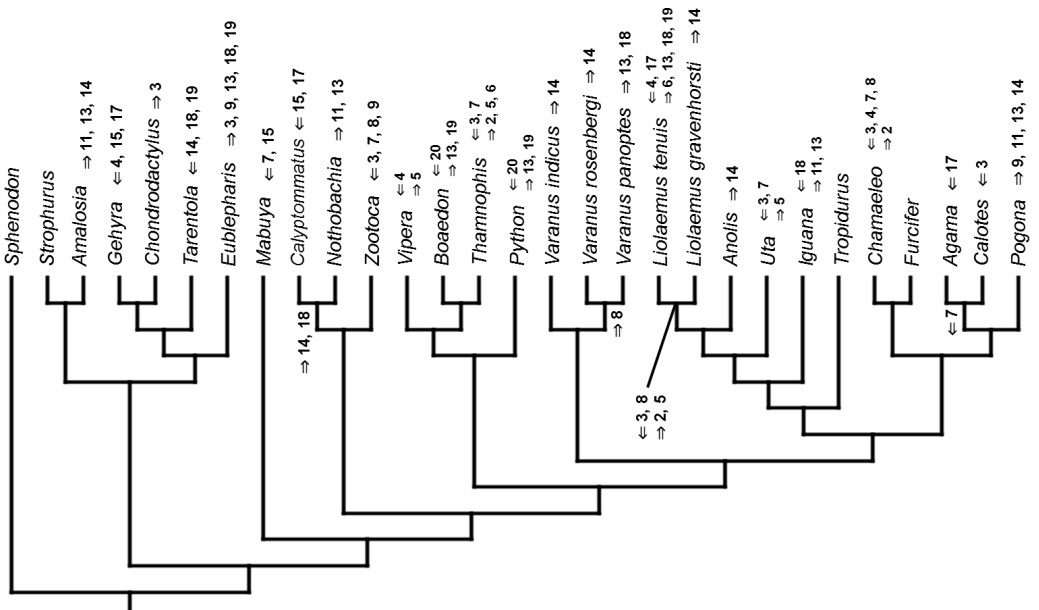

**Figure 3  Heterochronic events in lepidosaur evolution.** Mapped onto molecular phylogeny, using continuous data, in relation to the ancestral lepidosaur. Length of all branches equals 1. Numbers within boxes refer to developmental events (Table 2). Down arrow denotes earlier development of a given structure, while up arrow represents later development.

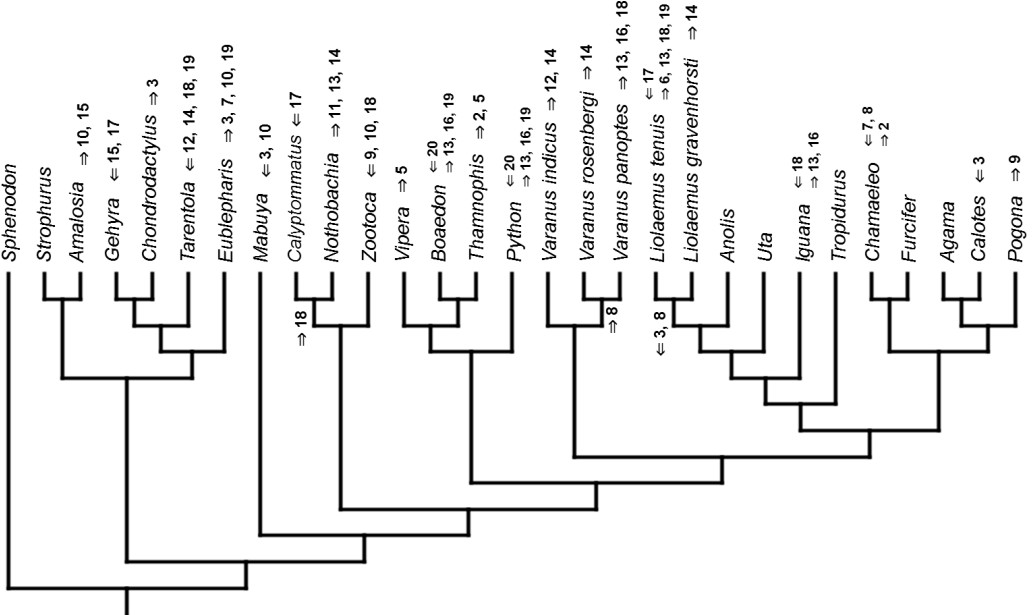

**Figure 4 Heterochronic events in lepidosaur evolution.** Mapped onto molecular phylogeny, using continuous data, in relation to the ancestral squamate. Length of all branches equals 1. Numbers refer to developmental events (Table 2). Down arrow denotes earlier development of a given structure, while up arrow represents later development.

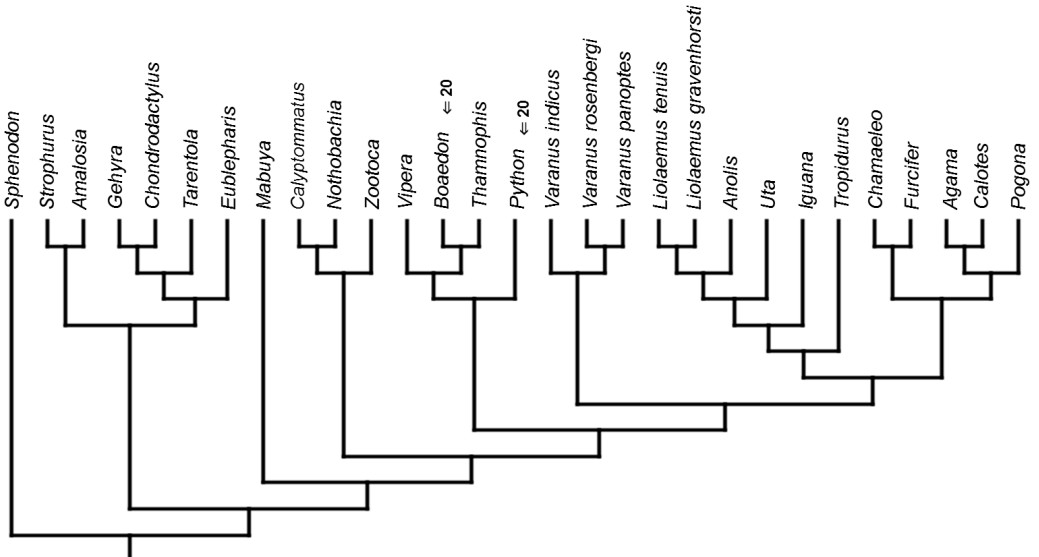

**Figure 5 Heterochronic events in lepidosaur evolution.** Mapped onto molecular, stratigraphically calibrated phylogeny, using continuous data, in relation to the ancestral lepidosaur. Numbers refer to developmental events (Table 2). Down arrow denotes earlier development of a given structure, while up arrow represents later development.

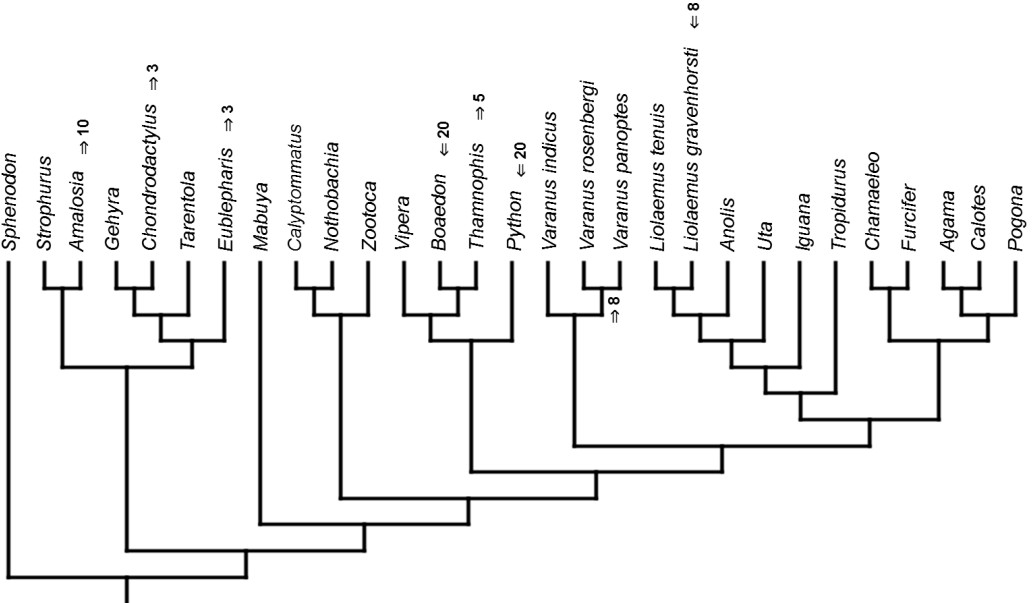

**Figure 6** **Heterochronic events in lepidosaur evolution.** Mapped onto molecular, stratigraphically calibrated phylogeny, using continuous data, in relation to the ancestral squamate. Numbers refer to developmental events (Table 2). Down arrow denotes earlier development of a given structure, while up arrow represents later development.

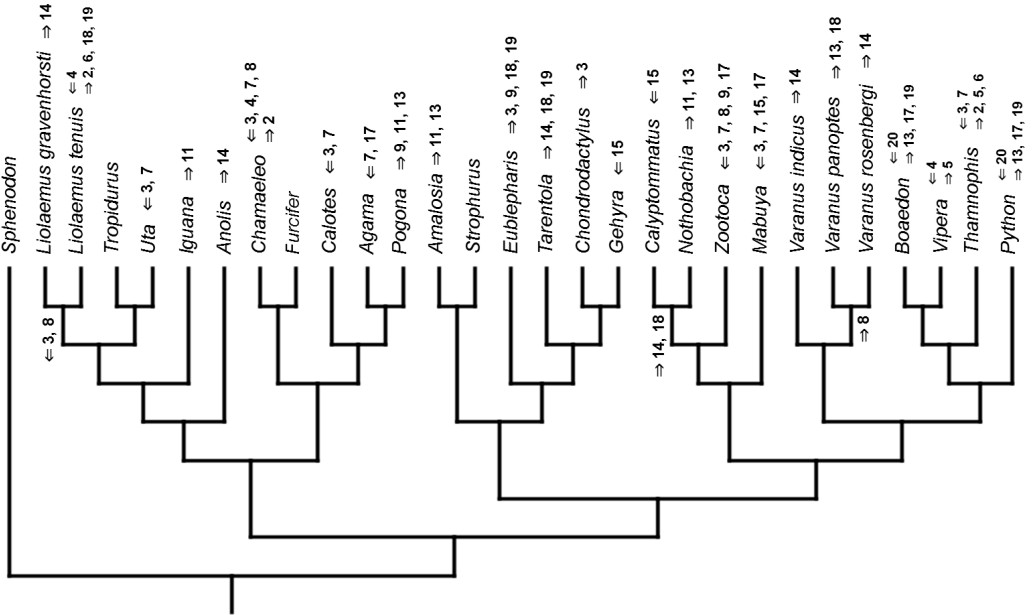

**Figure 7** **Heterochronic events in lepidosaur evolution.** Mapped onto morphological phylogeny, using continuous data, in relation to the ancestral lepidosaur. Length of all branches equals 1. Numbers refer to developmental events (Table 2). Down arrow denotes earlier development of a given structure, while up arrow represents later development.

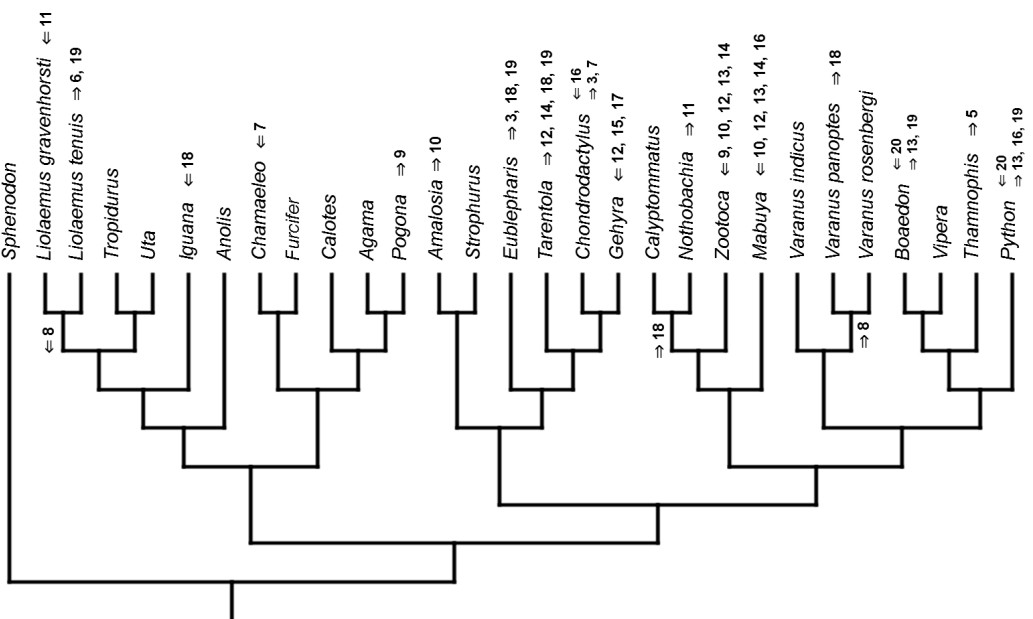

**Figure 8 Heterochronic events in lepidosaur evolution.** Mapped onto morphological phylogeny, using continuous data, in relation to the ancestral squamate. Length of all branches equals 1. Numbers refer to developmental events (Table 2). Down arrow denotes earlier development of a given structure, while up arrow represents later development.

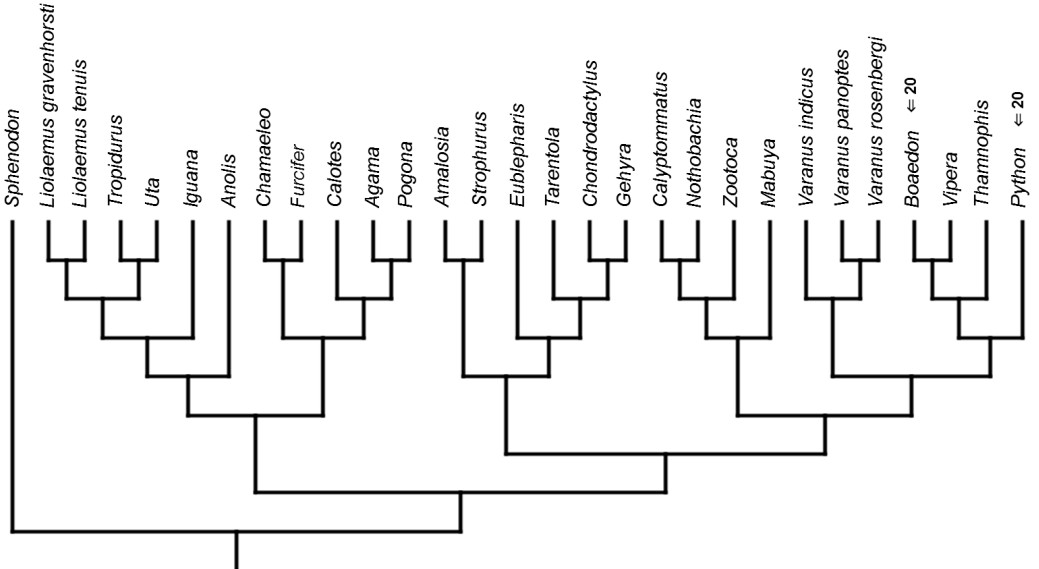

**Figure 9 Heterochronic events in lepidosaur evolution.** Mapped onto morphological, stratigraphically calibrated phylogeny, using continuous data, in relation to the ancestral lepidosaur. Numbers refer to developmental events (Table 2). Down arrow denotes earlier development of a given structure, while up arrow represents later development.

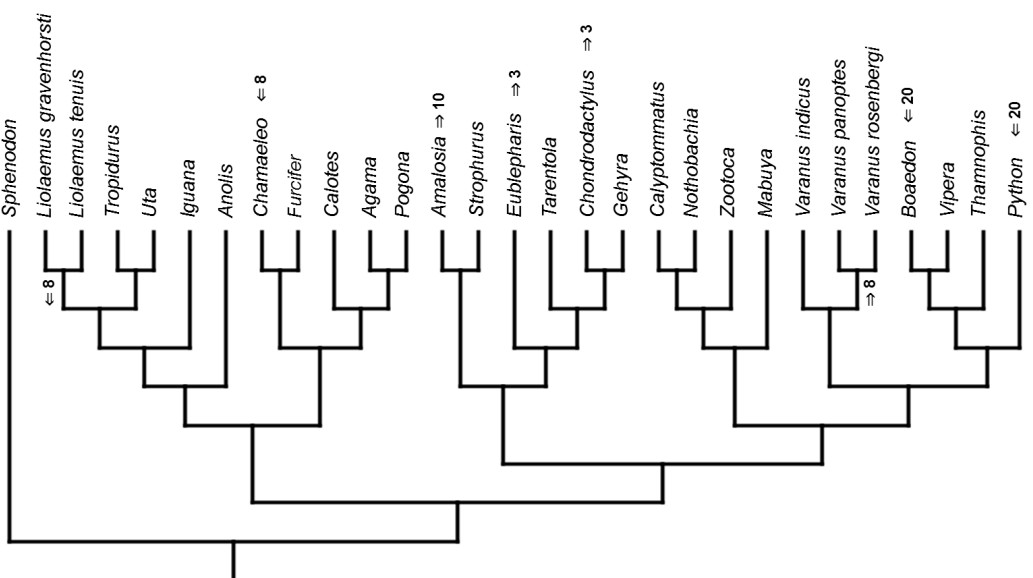

**Figure 10 Heterochronic events in lepidosaur evolution.** Mapped onto morphological, stratigraphically calibrated phylogeny, using continuous data, in relation to the ancestral squamate. Numbers refer to developmental events (Table 2). Down arrow denotes earlier development of a given structure, while up arrow represents later development.

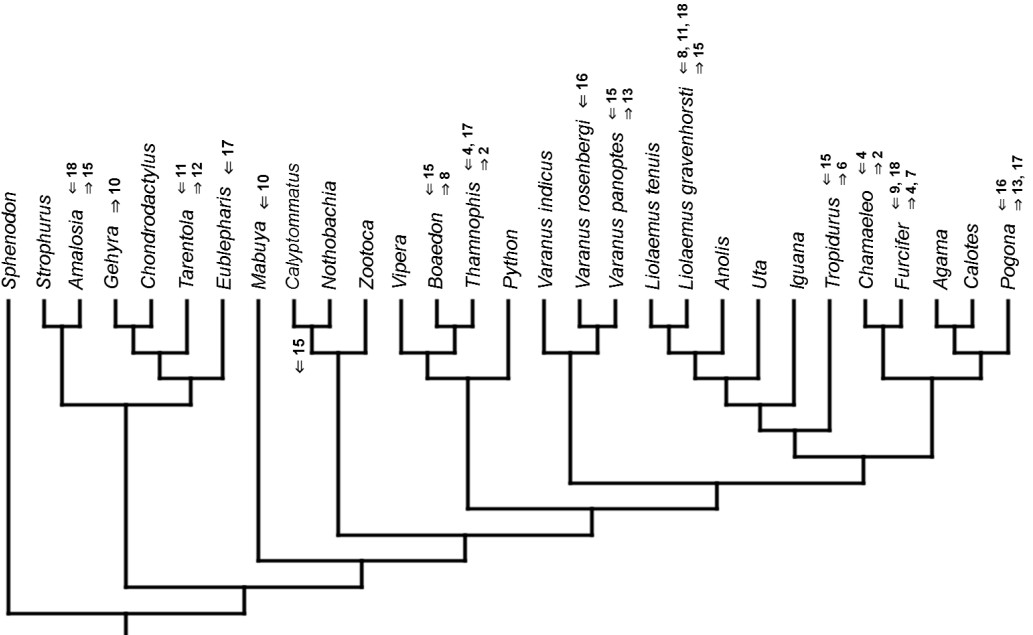

**Figure 11 Heterochronic events in lepidosaur evolution.** Mapped onto molecular phylogeny, using unordered event-paired characters. Numbers refer to developmental events (Table 2). Down arrow denotes earlier development of a given structure, while up arrow represents later development.

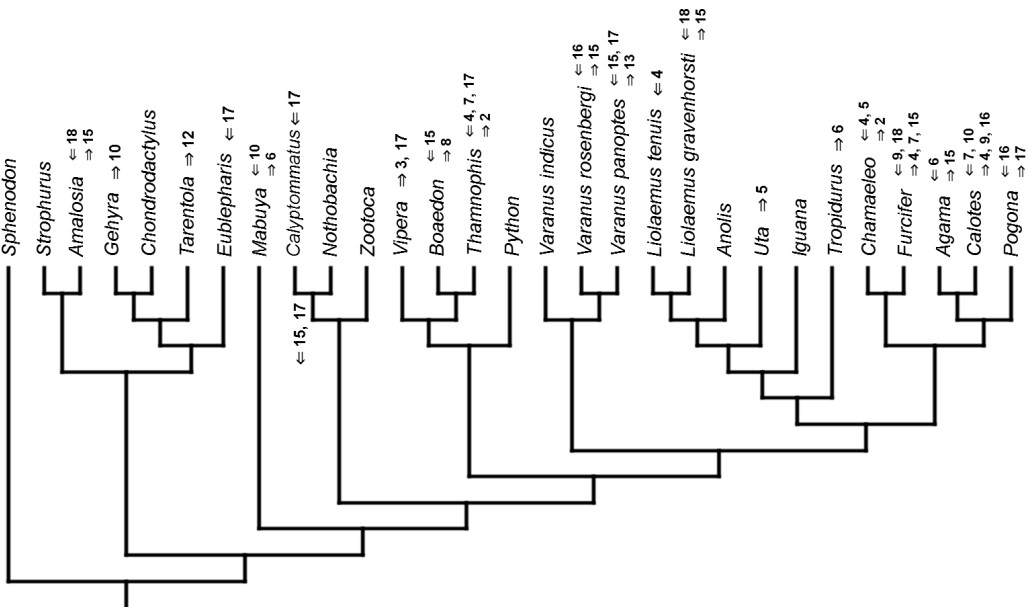

**Figure 12 Heterochronic events in lepidosaur evolution.** Mapped onto molecular phylogeny, using ordered event-paired characters. Numbers refer to developmental events (Table 2). Down arrow denotes earlier development of a given structure, while up arrow represents later development.

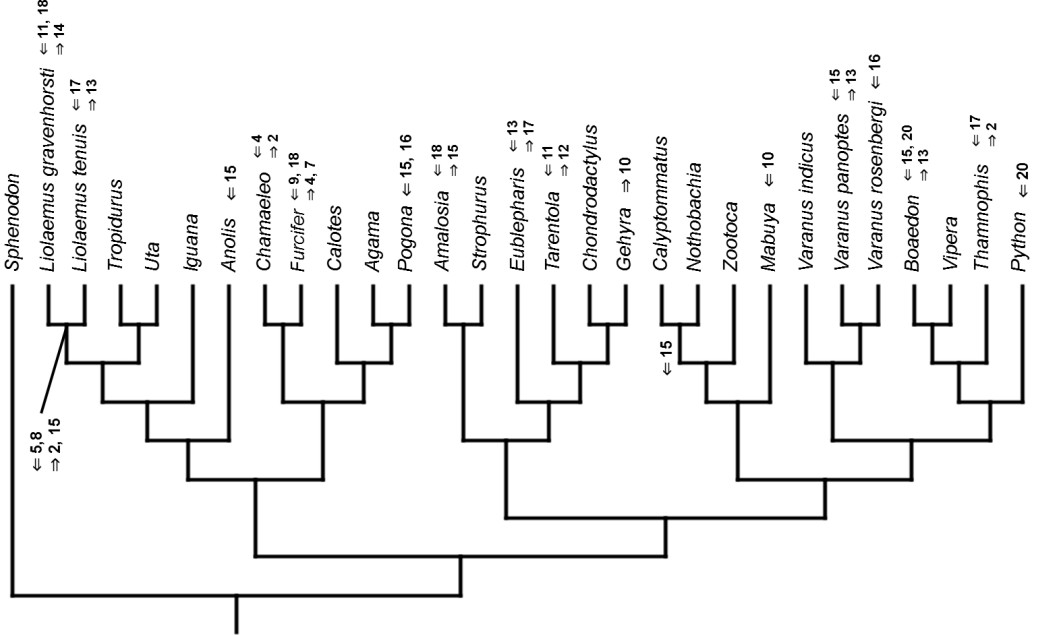

**Figure 13 Heterochronic events in lepidosaur evolution.** Mapped onto morphological phylogeny, using unordered event-paired characters. Numbers refer to developmental events (Table 2). Down arrow denotes earlier development of a given structure, while up arrow represents later development.

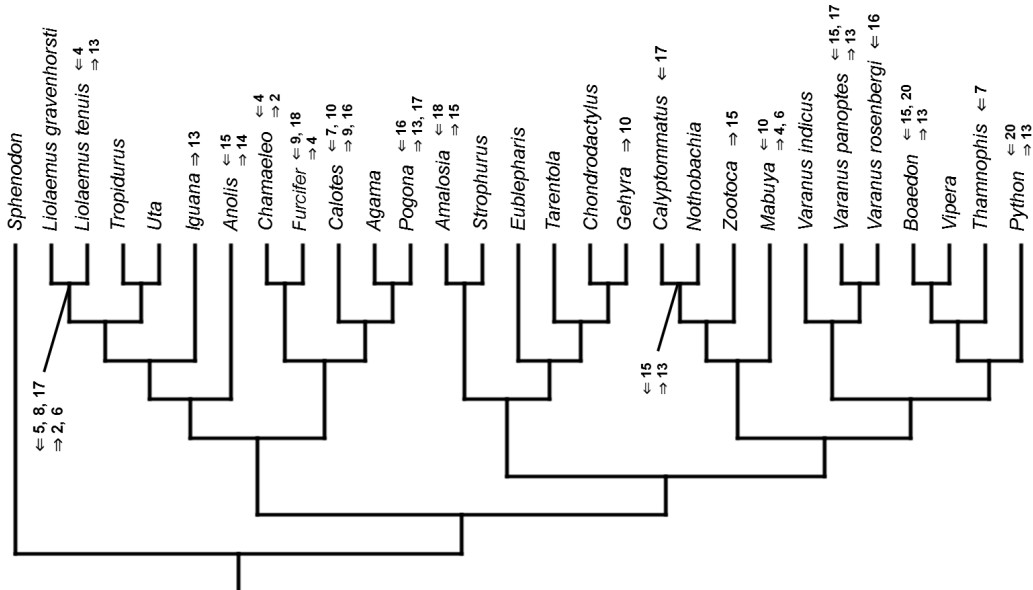

**Figure 14 Heterochronic events in lepidosaur evolution.** Mapped onto morphological phylogeny, using ordered event-paired characters. Numbers refer to developmental events (Table 2). Down arrow denotes earlier development of a given structure, while up arrow represents later development.

analysis—out of 28 included species, 11 are iguanians and six are gekkotans, while there are only three anguimorphs (and all of them belong to a single clade, *Varanus*) and one scincoid. Members of other important clades, like Amphisbaenia and Dibamidae, were not included. Some of these groups only recently were studied in terms of development (e.g., *Gregorovicova et al., 2012*). Moreover, development of lepidosaurs included in this analysis is incompletely known. Thorough study of developmental sequences of these and other members of these diverse clades will be beneficial to future analyses. However, it may be that homoplasies are very common in developmental sequences of squamates. Moreover, the phylogenetic signal in organogenetic events (at least those used in this study) may be weak or detectable only in deeper nodes of the phylogenetic tree (cf. *Jeffery et al., 2002b*; *Maisano, 2002*). This may be indicated by higher congruence between methods in reconstructing heterochronic events than between given phylogenies.

The only cladistic analyses that slightly resembled published phylogenies employed event-paired characters, especially ordered ones (Fig. 1B). In this analysis, eight of eleven included iguanian species formed a monophyletic group with *Thamnophis* and *Mabuya* that was sister to all other squamates. This resembles the morphological topology, where iguanians are sister group to all other squamates (e.g., *Estes, de Queiroz & Gauthier, 1988*; *Conrad, 2008*; *Gauthier et al., 2012*). This may suggest that developmental sequences of most iguanians and the tuatara are relatively similar. Under morphological topology, these similarities would represent symplesiomorphies but under molecular one, would be considered homoplasies. *Reeder et al. (2015)* suggested that support for basal placement of Iguania comes from the cranial characters. This is not the case in the present analysis. Character mapping and ancestral states reconstructions of event-paired data suggest that

potential symplesiomorphies between the tuatara and iguanians (as a whole or one of their major subgroups—Acrodonta and Pleurodonta) are connected with the relatively later torsion completion, rather than of some events concerning head development. Other groups recognized by morphological analyses also receive some support. For example, Scleroglossa are supported by earlier occurrence of torsion completion (simultaneous with occurrence of hyomandibular slit and allantois bud), unlike in tuatara and Iguania. Scincomorpha are supported by simultaneous development of otic placode, allantois bud and secondary optic vesicle.

Gekkotans differ from other squamates in later development of the allantois (*Andrews, Brandley & Greene, 2013*) but in that trait they resemble the tuatara. Under molecular topology, earlier development of the allantois bud supports the Unidentata (Table 4). This may represent a genuine signal of monophyly of that group, however, caution is warranted. Gekkotans display many paedomorphic features, including their morphology (e.g., *Daza, Bauer & Snively, 2014*) and development (*Jonasson, Russell & Vickaryous, 2012*). Thus, the condition in gekkotans may represent reversal to the primitive condition (presumably, as displayed by the tuatara) rather than plesiomorphy. This situation is similar to the development of a single egg tooth, which purportedly supports the monophyly of Unidentata (see discussion in *Assis & Rieppel, 2011*). To gain more insight into that matter, it would be crucial to sample development of dibamids, the only other non-unidentate squamates.

In the fossil time-calibrated continuous analysis, only one event in two species is inferred to show heterochrony in relation to the ancestral lepidosaur. This may seem surprising, as some squamates show heterochrony to the ancestral squamate (much closer phylogenetically). However, if all studied taxa are extant (as is the case in the present analysis), the long branches would result in wider confidence intervals and thus ancestral state reconstructions for deep nodes of the phylogenetic tree would be less certain (*Germain & Laurin, 2009*). Integration of data from fossils would be useful in that regard but it seems highly unlikely that information on organogenesis can be preserved in the fossil record, despite recent significant advances in developmental palaeobiology (e.g., *Skawiński &Tałanda, 2015*).

In the continuous analyses (both calibrated and uncalibrated and using either molecular or morphological topology), values of all developmental events of the tuatara are located within the confidence interval of the ancestral squamate. This suggests that present data are equally consistent with either hypothesis of squamate phylogeny (cf. *Germain & Laurin, 2009*).

In this study only two major phylogenetic hypotheses of squamates were used. It is not beyond imagination that neither of these phylogenies is fully correct. For example, in the analysis combining morphological and molecular data conducted by *Lee (2005)* the "fossorial group" is polyphyletic, as suggested by molecular analyses (e.g., *Wiens et al., 2012*; *Pyron, Burbrink & Wiens, 2013*), but division of squamates into Iguania and Scleroglossa is retained, as in morphological analyses (e.g., *Conrad, 2008*; *Gauthier et al., 2012*). This could, to some extent, explain the discrepancies in reconstructions of heterochronic events, as none of these would be done on the basis of the correct tree.

## CONCLUSIONS

Cladistic analyses conducted using characters generated by event-pairing and continuous analysis do not resemble any previously published phylogeny. Ancestral state reconstructions are equally consistent with both morphological and molecular hypotheses of squamate phylogeny. Results of the cladistic analyses, and the fact that reconstructions of heterochronic events show more similarities between certain methods than phylogenetic hypotheses, suggest that phylogenetic signal is at best weak in the studied developmental events.

## ACKNOWLEDGEMENTS

We acknowledge The Willi Hennig Society for making TNT freely available. Detailed and constructive comments made by Jessica A. Maisano and an anonymous referee greatly improved the manuscript. We thank John R. Hutchinson for editing.

### Funding

The authors received no funding for this work.

### Competing Interests

The authors declare there are no competing interests.

### Author Contributions

- Tomasz Skawiński conceived and designed the experiments, performed the experiments, analyzed the data, contributed reagents/materials/analysis tools, wrote the paper, prepared figures and/or tables, reviewed drafts of the paper.
- Bartosz Borczyk analyzed the data, wrote the paper, reviewed drafts of the paper.

### Data Availability

The raw data has been supplied as a Supplementary File.

### Supplemental Information

Supplemental information for this article can be found online at http://dx.doi.org/10.7717/peerj.3262#supplemental-information.

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
