# Peer review of "Evolution of developmental sequences in lepidosaurs"

_PeerJ, doi:10.7717/peerj.3262_

## Round 0.1 · original submission · Major Revisions

The reviewers agree that there must be much more thorough treatment of the available data in the literature for this analysis; not just relying on one study mainly. Please revise the MS accordingly and re-run the analyses- this will be required before it can be sent out for second review.

Also note the PDF document from Reviewer 2

Thanks for submitting to PeerJ.

Reviewer 1 ·

Basic reporting

no comment

Experimental design

no comment

Validity of the findings

no comment

Additional comments

This article represents a phylogenetic analysis based on developmental data. The authors have taken as a basis for their analysis the data previously presented by Adrews et al. (2013) and have added Sphenodon as an outgroup, what is definitely good for the analysis. Nevertheless I would recommend to authors also an adding of developmental sequences of squamates from other sources, and not only from Adrews et al. (2013). For example Naja (Jackson, 2002; Evans, Khanoon, 2014), Natrix(Korneva, 1969; Rupik, 2002), Iguana iguana (Lima, 2015), Varanus (Gregorovicova et al., 2012;Werneburg et al. 2015) Tarentola annularis (Khanoon, 2015) and many others. Adding of more data may probably lead to completely different results and conclusions. Moreover without new data this manuscript looks too alike with the previous paper of Adrews et al. (2013).

·

Basic reporting

Generally, this is a well-written article. The attached file details suggestions to improve grammar/clarity.

There is (at least) one very relevant paper that should be mentioned: Maisano, 2002 (citation in attachment). It applied event pair coding to ossification sequences in squamates....13 years before Werneburg and Sánchez-Villagra, 2015.

Experimental design

The authors do not add any new data (observations) to our knowledge base. This is unfortunate, as developmental data in squamates remain sparse. They only take data from pre-existing publications and analyze them in a different manner. However, this is not unusual in the scientific literature.

Validity of the findings

The conclusions reached by the authors are invalid unless they take into account ALL the relevant literature.

---

## Round 0.2 · Minor Revisions

We've received two reviews back and they are much more satisfied. I can conditionally accept the paper if minor revisions are done along the lines of reviewer 2. I agree with them that clade names generally are used in the singular, by most studies I've seen, but this is a relatively minor point so I leave this decision to the authors (with that nudge in the direction of what may be the majority) *as long as this is dealt with consistently throughout the paper*. Please make final revisions and submit. Thanks!!

Reviewer 1 ·

Basic reporting

no comment

Experimental design

no comment

Validity of the findings

no comment

·

Basic reporting

The authors have satisfactorily addressed my concerns from the first review.

The one part of their rebuttal with which I strongly disagree is treating clade names as plural. I could not find the Bauer, 2005 paper they cite in support of this. Even if it were appropriate, the authors are inconsistent in their application. In line 31 they write 'Squamata...are' (plural), and then in line 34, 'its evolutionary history' (singular). I guess this is a determination to be made by PeerJ editorial staff.

Other minor edits:

Line 130: insert ‘a’ before ‘conservative’
Line 351: should ‘University of’ be ‘Universidade de’?
Line 379: should ‘Py-Daniel’ come before ‘Pyron’?
Table 1: Rupik, 2002 is not in lit cite
Table 3: Sherratt is misspelled

Experimental design

No new comments from the first review.

Validity of the findings

The authors have addressed my concerns from the first review.

---

## Round 0.3 · accepted · Accept

Thank you for the attentive revisions- I am convinced that the paper is now acceptable!